



**Identification of high risk zones for geological origin hazards using**
**PALSAR-2 remote sensing data: Kelantan river basin, Peninsular**
**Malaysia**
Amin Beiranvand Pour *, Mazlan Hashim
Geoscience and Digital Earth Centre (Geo-DEC)
Research Institute for Sustainability and Environment (RISE)
Universiti Teknologi Malaysia, 81310 UTM JB, Malaysia
Abstract
Identification of high potential risk and susceptible zones for natural hazards of geological
origin is one of the most important applications of advanced remote sensing technology. In
this study, the recently launched Phased Array type L-band Synthetic Aperture Radar-2
(PALSAR-2) onboard the Advanced Land Observing Satellite-2 (ALOS-2), remote sensing
data were used to map geologic structural and topographical features in the Kelantan river
basin for identification of high potential risk and susceptible zones for landslides and
flooding areas. The data were processed for comprehensive analysis of major geological
structures and detailed characterizations of lineaments, drainage patterns and lithology at both
regional and district scales. Red-Green-Blue (RGB) colour-composite was applied to
different polarization channels of PALSAR-2 data to extract variety of geological
information. Directional convolution filters were applied to the data for identifying linear
features in particular directions and edge enhancement in the spatial domain. Results indicate
that lineament occurrence at regional scale was mainly linked to the N-S trending of the
Bentong-Raub Suture Zone (BRSZ) in the west and Lebir Fault Zone in the east of the
Kelantan state. Combination of different polarization channels produced image maps contain
important information related to water bodies, wetlands and lithological units. The N-S, NE-
SW and NNE-SSW lineament trends and dendritic, sub-dendritic and rectangular drainage
patterns were detected in the Kelantan river basin. The analysis of field investigations data
indicate that many of flooded areas were associated with high potential risk zones for hydro-



geological hazards such as wetlands, urban areas, floodplain scroll, meander bend, dendritic
and sub-dendritic drainage patterns, which are located in flat topography regions. Numerous
landslide points were located in rectangular drainage system that associated with topographic
slope of metamorphic and quaternary rock units. Consequently, structural and topographical
geology maps were produced for Kelantan river basin using PALSAR-2 data, which could be
broadly applicable for landslide hazard mapping and identification of high potential risk zone
for hydro-geological hazards. Geo-hazard mitigation programmes could be conducted in the
landslide recurrence regions and flooded areas for reducing natural catastrophes leading to
loss of financial investments and death in the Kelantan river basin. In this investigation,
PALSAR-2 has proven to be successful advanced earth observation satellite data for disasters
monitoring in tropical environments.
Key words: ALOS-2; PALSAR-2; Geological origin hazards; Tropical environments; Peninsular Malaysia
*Corresponding Author. Tel: +607 -5530666; Fax: +607- 5531174; Email address:
beiranvand.amin80@gmail.com; a.beiranvand@utm.my; mazlanhashim@utm.my

## 1. Introduction

Advances in remote sensing technology allow the application of Synthetic Aperture Radar
(SAR) data in geological structural analysis for tropical environments (Shimada and Isoguchi,
2002; Morelli and Piana, 2006; Ramli et al., 2009; Pour and Hashim, 2013, 2014a,b,
2015a,b). Structural field mapping is often difficult in heavily vegetated Terrain. This is the
case with the study area where dense vegetation cover, deep weathering and scarcity of
bedrock exposure hampers geological structure mapping over a long distance. In addition, the
use of optical remote sensing data is limited due to the persistent cloud coverage of the study
area for most part of the year (Ramli et al., 2009; Hashim et al., 2013). SAR data contain
potential to penetrate cloud and vegetation and unlike the optical data are dependent upon the
surface roughness of the materials, ideally suited to mapping lineaments (faults and fractures)
in tropical environments. Lineaments are related to large structural fractures, which represent
zones of weakness in the brittle part of the lithosphere. The presence of linear tectonic
structures is one of the important factors in geological hazard occurrences (Bannert, 2000a,b).
Lineaments are represented by faults (linear features), lithological contacts between rock
units and drainage patterns in any area (van der Pluijm and Marshak, 1997). Observation
from satellite images (Landsat Thematic Mapper) in the Himalayan Mountains of Nepal and
China revealed a clear connection between of active faults, associated with earthquakes and
the occurrence of large landslides (Bannert, 2000a,b). Therefore, delineation of faults and
fractures using advanced remote sensing technology in any region is necessity to assess the
potential for many natural geological hazards. Numerous investigations have used faults,
drainage patterns and lithology factors as important factors to measure and map the
susceptibility of geological hazard (Guzzetti et al., 1999; Dai and Lee, 2002; Suzen and
Doyuran, 2004; Abdullah et al., 2013).
The Advanced Land Observing Satellite-2 (ALOS-2) was launched on May 24, 2014 as
successor of ALOS-1 (launched on January 24, 2006 and decommissioned in May 2011)
(http://global.jaxa.jp/press/2014/05/20140524_daichi2.html; Blau, 2014). The ALOS-2 is
exclusively installed with the Phased Array type L-band Synthetic Aperture Radar-2
(PALSAR-2) using microwaves to maximized its ability compare to the ALOS-1, on which
three sensors (two optical and one microwave devices) were onboard (Igarashi, 2001; Suzuki
et al., 2012). In particular, L-band microwave from PALSAR-2 has ability to penetrate
vegetation due to relatively long wavelengths (about 24 cm), making the data particularly
useful for geological structural mapping in tropical environments (Igarashi, 2001; Rosenqvist
et al., 2004; ERSDAC, 2006; Arikawa et al., 2010; Yamamoto et al., 2013; Pour and Hashim,
2013, 2014 a,b, 2015a,b; Shimada et al., 2015). The wavelength of the L-band is relatively
long among microwaves (C-band: about 6 cm and X- band: about 3 cm), allowing it to travel
all the way down to the ground through vegetation (Woodhouse, 2006). Not only can
information be, obtained about vegetation but information of the ground surface can be
obtained as well. Additionally, L-band is less affected by the growth of vegetation, which is
useful for SAR interference analysis (Interferometry). Therefore, L-band is capable to acquire
changes on the land more precisely compared to shorter wavelength SAR when some
diastrophism takes place due to an earthquake or a volcanic activity and floods or landslides
caused by a natural disaster (Suzuki, 2014). Accordingly, a further increase in amount of
information for geological structural mapping could be derived from the recently launched
PALSAR-2 data. Analysis of the data can provide completely new insights into heavily
vegetated areas threatened by natural hazards of geological origin. Consequently, the
advanced SAR remote sensing data are broadly applicable for geo-environmental research to
identify the causes of natural disasters and point the way to rehabilitation measures especially
in tropical environments. To date, few studies used L-band SAR remote sensing data for
geological structure mapping in tropical environments (Pour and Hashim, 2013, 2014a,b,
2015a,b). This study is the first time that L-band SAR remote sensing data is used for
identification of high potential risk and susceptible zones for natural hazards of geological
origin in tropical environments. It is dire need to apply this approach in Malaysia and other
parts of South East Asia that have inaccessible regions and high potential zones for natural
hazards of geological origin hidden by dense rainforest.
Yearly, several landslides occur during heavy monsoon rainfall in Kelantan river basin,
Peninsular Malaysia, which are obviously connected to geological structures and
topographical features of the region. In recent years especially, there have been many severe
flooding events (in the year 2005, 2006, 2007, 2008, 2009, 2014 and 2015) which have led to
significant damage to livestock, agricultural produce, homes and businesses in the Kelantan



river basin (Pradhan, 2009; Pradhan et al., 2009; Pradhan and Youssef, 2011; Tehrany et al.,
2013; Nazaruddin, et al., 2014). The problem stems from the inappropriate use of lands that
are vulnerable to erosion, quick water runoff and slope failure. Recent challenge is to identify
high potential risk and susceptible zones for natural hazards of geological origin in the
Kelantan river basin using advanced remote sensing technology. Additionally, accurate and
up-dated geological structure and topographical maps are largely lacking for the Kelantan
river basin. Therefore, the objectives of this study are (i) to identify high potential risk and
susceptible zones for geological origin hazards using the recently launched ALOS-2-Phased
Array type L-band Synthetic Aperture Radar-2 (PALSAR-2) remote sensing data in the
Kelantan river basin at regional and district scales; (ii) to produce accurate geological
structure and topographical maps for the Kelantan river basin using PALSAR-2 data; and (iii)
to compare the detected high potential risk and susceptible zones with high damaged areas in
recent flooding events in the Kelantan river basin.

## 2. Study area

Peninsular Malaysia is composed of central segment of Southeast Asian continental core of
Sundaland (Metcalfe, 2013a,b). The state of Kelantan is located in north-eastern corner of
Peninsular Malaysia (Fig. 1). Kelantan river is the major river in the region. It appears at the
convergence of the Galas river and Lebir river near Kuala Kari and meanders over the coastal
plain until it finally degrades into the South China Sea. Kelantan river basin covers 923 km$^2$,
which is about 85% of the Kelantan state's surface area. It is composed of flat slope to
moderately sloping areas in northern part and steep scraps and high slopes in the southern
part of the river basin (Pradhan et al., 2009). A wide variety of rocks consisting of igneous,
sedimentary and metamorphic rocks are distributed in a north-south trend in the Kelantan

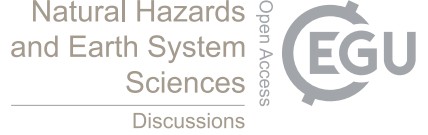

state. Typically, four types of rocks are classified in the region, including granitic rocks,
sedimentary/metasedimentary rocks, extrusive rocks (volcanic rocks) and unconsolidated
sediments (Fig. 2). Localised geological features comprise faulting and jointing in the granitic
rocks and folding, faulting and jointing in the sedimentary rocks. Granitic rocks are
distributed in the west (the Main Range granite) and east borders (the Boundary Range
granite) of the state of Kelantan (Rahman and Mohamed, 2001; Department of Minerals and
Geoscience Malaysia, 2003; Heng et al., 2006).
The Main Range granite is located in the west of the state, which is stretched along western
Kelantan up to the boundary of Perak and Pahang states and Thailand boundary (Fig. 2). The
dominant structural trend in Kelantan is along N-S to NW-SE direction that derived from
postorogenic phase (Ghani, 2009). The Main Range granite is located along the western
margin of the Bentong-Raub Suture Zone (BRSZ) and extends north to Thailand (Schwartz et
al., 1995; Metcalfe, 2000). The north-south trending Bentong-Raub Suture (approximately 13
km wide) extends from Thailand through Raub and Bentong to the east of Malacca,
Peninsular Malaysia. The BRSZ is characterized by a series of parallel topographic north-
south trending lineaments and the presence of small bodies of mafic to ultramafic rocks that
are commonly serpentinized (Tan, 1996).  The Lebir Fault Zone is located in the eastern part
of Kelantan state (Fig. 2), which is one of the major lineaments in Peninsular Malaysia and
considered to be post-Cretaceous and sinistral strike-slip fault (Tjia, 1989; Harun, 2002).
The landscape in the state of Kelantan has been divided into four types, including
mountainous areas, hilly areas, plain areas and coastal areas (Unjah et al., 2001; Raj, 2009).
Mount Chama (Gunung) is the highest point (2171m) in the Kelantan state, which is located
in Gua Musang district in the western part of the state, near the border of Perak State
(Nazaruddin, et al., 2014). Quaternary deposits consist of alluvium deposits or



unconsolidated sediments. Triassic marine siliciclastics, volcaniclastics, sandstone and
limestone are the other sedimentary rocks occur in the Kelantan state.


**3. Materials**
PALSAR-2 of the ALOS-2 has been significantly improved from the ALOS-1's PALSAR in
all aspects, including resolution, observation band and time lag for data provision (Arikawa et
al., 2010; Kankaku et al., 2010; Suzuki et al., 2012). ALOS-2 science proficiencies contain
global environmental monitoring using the time-series PALSAR-2. The research objective
encompasses biospheric, cryospheric, coastal ocean research and disaster monitoring and
mitigation (Shimada, 2013). PALSAR-2 is a microwave sensor that emits L-band radio
waves and receives their reflection from the ground to acquire information (Suzuki et al.,
2012). It has three observation modes, including (i) Spotlight mode: the most detailed
observation mode with 1 by 3 meters resolution and observation width of 25 km; (ii) Strip
map mode: a high-resolution mode with the choice of 3 (ultra fine), 6 (high sensitivity) or 10
(fine) meters resolution and observation width of 50 or 70 km; and (iii) ScanSAR mode: a
broad area observation width of 350 (nominal) or 490 (wide) km and resolution of 100 or 60
meters (Yamamoto et al., 2013; Shimada et al., 2015).
Selection of the most suitable observation mode and time acquisition of PALSAR-2 data for
the research objectives will maximize the efficiency of geo-environmental monitoring works.
In this investigation, a ScanSAR mode dual polarization (level 3.1) and two Fine mode dual
polarization (level 3.1) PALSAR-2 scenes were obtained from ALOS-2 data distribution
consortium online system Remote Sensing Technology Center of Japan (RESTEC)
(http://www.restec.or.jp/english/index.html) and PASCO Corporation (http://en.alos-

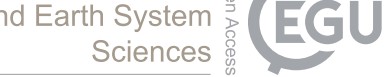

pasco.com; https://satpf.jp/) for comprehensive analysis of major geological structures and
detailed characterizations of lineaments in the state of Kelantan. The data used in this study
were acquired during dry season (June to August, 2015). Two distinct wet seasons from
September to December and February to May are reported by Malaysian Meteorological
Department (MMD) in Peninsular Malaysia (Mardiana et al., 2000). Precipitation and soil
moisture variations have influence on detailed geologic lineament analysis using microwave
signal. SAR data are highly sensitive to water content in the soil because of large contrast
between dielectric properties of water and dry soil (Eagleman and Lin, 1976; Jackson and
Schmugge, 1989; Lacava et al., 2005). Hence, SAR data acquired during dry seasons contain
more useful information for detailed geological structural mapping in tropical environments.
In this study, the ScanSAR observation mode has 60 m resolution and 490 km swath width
(wide mode), and the fine observation mode contains data with 10 m resolution and 70 km
swath width. Dual polarization for the both data includes HH+HV polarization images (HH=
Horizontally transmitted and Horizontally received, HV= Horizontally transmitted and
Vertically received). HV channel is a channel that transmits a vibrating wave in a horizontal
direction against the ground (H) and receives it vertically (V). It is expected to acquire more
detailed observation data on ground. Particularly, HV channel (cross-polarization) in the
ScanSAR and fine observation modes increase the amount of information extraction for
geological structural mapping at both regional and districts scales (Pour and Hashim,
2015a,b). HV polarization is more suitable for lineament extraction and edge enhancement in
tropical environments than other polarization channels, because cross-polarization is more
sensitive to lineament and also enhances penetration (Henderson and Lewis 1998; Pour and
Hashim, 2015a,b). Penetration is proportional to wave-length, and cross-polarization also
enhances penetration (Henderson and Lewis 1998). Therefore, HV polarization channel
records more geological features that cover by dense vegetation. In the Level 3.1 product of



PALSAR-2, image quality corrections (noise reduction and dynamic range compression) are
performed to level 1.5 of PALSAR-2 data. It should be noted that level 1.5 of PALSAR-2
data contain the following characteristics (i) range and multi-look azimuth compressed data is
represented by amplitude data; (ii) range coordinate is converted from slant range to ground
range; and (iii) map projection is also performed (Japan Aerospace Exploration Agency,
2014). The positioning accuracy of PALSAR-2 data is excellent and it does not always
require the field survey to modify the positioning accuracy. So, it can reduce the map
production time and contribute to cost reduction as well (Shimada et al., 2015). Subsequently,
the data used in this study are geo-referenced. The data were processed using the ENVI
(Environment for Visualizing Images) version 5.2 software package.


**4. Methods**

Lineaments in remote sensing images have topographic relief and are often a surface
expression of 3D geological structures in the subsurface. Lineaments, associated with brittle
and ductile deformation zones appear as rectilinear and curvilinear patterns in remotely
sensed images. Detailed extraction of lineaments helps in reconstructing the tectonic history
of a region (Clark and Wilson, 1994). Geological lineament features are attributed to paleo-
tectonic and /or neo-tectonic activity of a region. The use of remote sensing data in
delineation tectonically significant lineaments has been demonstrated in many geological
settings (Raharimahefa and Kusky, 2007, 2009; Amri et al., 2011; Hashim et al., 2013;
Hamimi et al., 2014; Pour and Hashim, 2014a,b, 2015a,b).
Spatial transforms provide reliable and robust image processing techniques to extract the
spatial information from remote sensing data. The techniques help to maximize clarity,



sharpness and details of features of interest towards information extraction and further
analysis. Spatial convolution filtering is based primarily on the use of convolution masks.
This flexibility makes convolution one of the most useful tools in image processing. The
procedure could be used to enhance low and high frequency details, as well as edges in the
imagery (Haralick et al., 1987; Jensen, 2005; Schowengerdt, 2007). Linear features are
formed by edges in remotely sensed images. Some linear features occur as narrow lines
against a background of contrasting brightness; others are the linear contact between adjacent
areas of different brightness. Edge enhancement delineates these edges and makes the shapes
and details comprising the image more conspicuous and easier to analyze (Jensen, 2005). It
can be used in geological applications to highlight rectilinear and curvilinear patterns in
remote sensing images.
Systematic image processing techniques were implemented to the PALSAR-2 data for
geological structures and lineament mapping at both regional and district scales in the state of
Kelantan. It is necessary to treat the speckle in radar images by filtering before it can be used
in various applications (Sheng and Xia, 1996). The presence of speckle in radar images
reduce the detectability of ground targets, obscures the spatial patterns of surface features and
decreases the accuracy of automated image classification (Lee and Jurkevich, 1994;
Sveinsson and Benediktsson, 1996). However, image quality corrections (noise reduction)
have been already applied to Level 3.1 product of PALSAR-2, but some speckles (salt and
pepper noise) could be still seen in the images. In this study, the median spatial convolution
filter was used for noise removal and smoothing the PALSAR-2 images. The median filter is
a particularly useful statistical filter in the spatial domain, which effectively remove speckle
(salt and pepper noise) in radar images without eliminating fine details (Russ, 2002; Research
Systems, Inc., 2008). The median operation has the effect of excluding pixels that do not fit
the typical statistics of the local neighborhood, i.e., outliers. Isolated noise pixels can



therefore be removed with a median filter (Schowengerdt, 2007). The median filter is
especially useful for removing shot noise (pixel values with no relation to the image scene)
and has certain advantages such as it does not shift boundaries and has the minimal
degradation to edges (Eliason and McEwen, 1990; Russ, 2002; Jensen, 2005). In this study,
3*3 neighborhood convolution mask (kernel) was applied to the PALSAR images. Image
Add Back value was entered 60%. The Image Add Back value is the percentage of the
original image that is included in the final output image. This part of the original image
preserves the spatial context and is typically done to sharpen an image. The directional nature
of geological lineaments accentuates the need for directional filtering to obtain maximum
structural mapping efficacy. Edge enhancing filter highlights any changes of gradient within
the image features such as structural lines (Carr, 1995; Sabins, 1996; Vincent, 1997; Tripathi
and Gokhale, 2000; Research Systems, Inc., 2008).
In this study, for identifying linear features in particular directions and edge enhancement in
the spatial domain, directional convolution filters were applied to the median resultant image.
Directional filtering technique is a straightforward method for extracting edges in the spatial
domain that approximates the first derivative between two adjacent pixels. The algorithm
produces the first difference of the image input in the horizontal, vertical, and diagonal
directions (Haralick et al., 1987; Carr, 1995; Sabins, 1996; Vincent, 1997; Jensen, 2005). As
a result, many additional edges of diverse orientations are enhanced (Richards and Jia, 1999).
The edges appear as a plastic shaded-relief format (embossing) in the image because of the 3-
D impression conveyed by the filtering (Schowengerdt, 2007). Directional filter is used for
producing artificial effects suggesting tectonically controlled linear features (Pour and
Hashim, 2015a,b).
Directional filters were used to enhance specific linear trends in the median resultant image.
Four principal directional filters: N-S, E-W, NE-SW, and NW-SE with 5*5 and 7*7 kernel

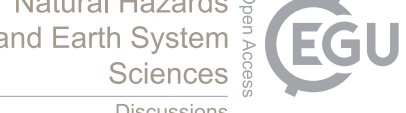

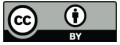

sizes were applied to ScanSAR and Fine scenes, respectively. 5*5 kernel matrix was selected
for ScanSAR scene to enhance rough/smooth and semi-rough features at regional scale in
northern part of Peninsular Malaysia (Table 1). 7*7 kernel matrix was applied to Fine scenes
for enhancing semi-smooth and smooth/rough features at district scale in the Kelantan state
(Table 2) (Chavez and Bauer, 1982; Jensen, 2005). Directional filter angles were adjusted as
N-S: $0°$, E-W: $90°$, NE-SW: $45°$, and NW-SE: $135°$. North (up) is zero degrees and the other
angles are measured in the counterclockwise direction. Image Add Back value was entered
60%. Interpretation and analysis of geological features and enhanced lineaments was
accomplished using systematic remote sensing techniques. Red-Green-Blue (RGB) colour-
composite was used for different polarization configuration of PALSAR-2 data to provide
visual interpretation of geological structures and lithological units in the study area. Image
processing results were compared with the geological and general topography maps of
Kelantan state (1:100,000 scale) (Department of Minerals and Geoscience Malaysia, 2003).
Furthermore, for verification of the image processing results, fieldwork was conducted during
a scientific expedition in Kelantan river basin between 20 and 25 June 2015, to collect data in
landslide affected zones and flooded area by Global Positioning System (GPS) surveying,
rock sampling and photographs recording. GPS survey was carried out using a Garmin®
MONTERRA® with an average accuracy 5 m in 453 landslide affected location points in the
study area.








## 5. Results and discussion

### 5.1 lineament extraction and lithology discrimination at regional scale

A wide-swath ScanSAR observation mode of PALSAR-2 was used for comprehensive analysis of major geological structures, which shows mega-geomorphology and mega-lineaments in the Kelantan state. Figure 3 shows RGB colour-composite of HH polarization channel in red, HV polarization channel in green and HH+HV polarization channel in blue for the ScanSAR median resultant image. The RGB colour-composite yields an image with great structural details and geomorphological information. The different colors in the image indicate different backscattering signals from the ground. The Main Range granites located in in the western part of the image and the Boundary Range granite in the east borders of Kelantan state appear as light green to green in colour, which shows the regions with high altitude in the scene (Fig. 3). Major transcrustal lineaments such the Bentong-Raub Suture Zone (BRSZ) and Lebir Fault Zone are also detected. Quarternary deposits, Triassic marine siliciclastics, volcaniclastics, sandstone and limestone are manifested as pink to purple tones consisting of lands with low elevation (Fig. 3). Lakes and main river systems are portrayed blue to dark blue in the image. In fact, the lines formed by blue to dark blue colour on the image are faults and fracture zones occupied by streams. The river pattern is one of the most important factors contributing to the lineament because it reflects the nature of the existing fracture system (Suzen and Toprak, 1998).

Figure 4 shows ScanSAR HV polarization image that is superimposed by general topography map of the Kelantan state (Department of Minerals and Geoscience Malaysia, 2003). It is evident that the morphology of the study area is largely controlled by rock type and structure. High elevation areas (500-100 m and <1000 m) in the Kelantan state are mountainous areas associated with Main Range granites (in the west) and Boundary Range granite (in the east),




which are detected as green to light green colour in the Figure 3. Hilly, plain and coastal
areas with altitude between 500 to 50 m are associated with sedimentary rocks, which are
manifested pink to purple colour in the Figure 3.
Figure 5 shows the resultant image map for N-S, NE-SW, and NW-SE (R: 0°, G: 45°, B: 135°)
directional filters.  Major change in deformation style is obvious from the west to the east in
Figure 5. Structural analysis reveals four distinct parts from the west to the east, including (i)
western part of the scene by ductile fabrics; (ii) western of the BRSZ affected mainly by
brittle deformation; (iii) ductile-brittle deformation between the BRSZ and Lebir Fault Zone;
and (iv) brittle-ductile fabrics between Lebir Fault Zone and eastern coastal line. Lineament
occurrence in Figure 5 is mainly linked to the N-S trending of the BRSZ and Lebir Fault
Zone. Generally, major faults are strike-slip with both dextral and sinistral movements, which
trend N-S and NW-SE. NW-SE trending strike-slip faults moved sinistrally in the Lebir Fault
Zone. The sinistral movement along the Lebir Fault Zone is responsible to the formation of
folding and reverse faulting adjacent to the fault and surrounding area. These structures
characterized transpressive tectonic regime in the Peninsular Malaysia (Richter et al., 1999;
Harun, 2002).
The collision zone and compressional structures appear clearly in the west of the BRSZ in
Main Range granites (Fig. 5). Deformation in this region shows the shortening zone oriented
parallel to the BRSZ. Several faults, joints and fractures represent brittle deformation events
in the region that mostly strike NW-SE. Generally, most of the short lineaments are clustered
in the collision zone. Ductile deformation in the western margin of the image (Fig 5) includes
upright asymmetrical mega folds with axial surfaces oriented W-E. Hence, the contractional
strain direction affected this domain is from NE-SW followed by dextral shearing. The
deformed area zone between the BRSZ and Lebir Fault Zone represents faults system and
folded area. NW-SE striking strike-slip faults and NE-SW normal faults are dominated brittle





structural elements within this domain. Ductile deformation is demonstrated by several open
upright folds, which have W-E to NE-SW axial plans (Fig. 5). Brittle-ductile fabrics in the
eastern part of image between Lebir Fault Zone and eastern coastal line illustrate curved
shear zone that occupied by several N-S and NW-SE striking faults, fractures and joints. A
mega concentric fold surrounds the shear zone with a WE striking axial surface. According to
the orientation of the lineaments, sinistral movement along the Lebir Fault Zone is generated
the tectonic features. Some N-S and NE-SW trending normal faults and small curvatures are
also identifiable near the eastern coastal line (Fig. 5).

*5.2 Lithology mapping at district scale*

Figure 6 shows fine mode merged image of the northern and southern parts of the Kelantan
state. In this figure, HH polarization channel was assigned to red colour, HV polarization
channel to green colour and HH+HV polarization channels to blue colour for generating an
RGB image for the study area. This colour combination produced an image map contains
important information related to water bodies, wetlands and geological structural features.
River systems and lakes appear black especially in north-eastern part of the image and
wetlands as mauve colour (Fig. 6). Smooth surfaces such as calm water bodies appear dark in
SAR images due to reflection of radar signal. Hence, no returning radar signal could be
detected in receiving antenna (Thurmond et al., 2006; Pour and Hashim, 2015a). Besides,
best soil moisture/wetness information is achievable from L-band microwave because it
comes out from deeper soils. In fact, vegetation is almost transparent and roughness effects
are negligible using L-band microwave for soil wetness mapping/monitoring (Jackson and
Schmugge 1989; Lacava et al., 2005). Soil moisture is one the most important factors in
hydro-geological hazards, especially since the soil response is affected by its status of



saturation. Accordingly, the identification of wetlands is very important for flood forecasting
and prevention in the state of Kelantan. Wetlands are more distributed in northern part of the
study area (Fig. 6). A vast wetland (strong mauve colour) represents clearly in the central
north segment of the image. Several wetlands as light to strong mauve colour are observable
in central east, south and south-western part of the image (Fig. 6).
The RGB image map (Fig. 6) was compared with the geological map of the Kelantan state. It
is evident that most of the detected wetlands are associated with sedimentary and
metamorphic rocks. A few of the wetlands are located in the granites bedrock areas in the
west (the Main Range granite) and east (the Boundary Range granite) of the state of
Kelantan. The granites bedrock areas are all characterized by dissected hilly to mountainous
terrain that gives rise to isolated highlands and mountain ranges (Fig. 6). These granitic areas
represent the outcrops of a number of granite batholiths that are generally elongated along a
N-S to NNW-SSE direction (Bignell and Snelling, 1977).
The amplitude of the radar signal is highly sensitive to the physical properties of the ground
surface, which produces the brightness of the surface to vary much more than optical or
infrared images (Robinson et al., 1999). Hence, combination of different polarization
channels of fine mode PALSAR-2 data contains considerable information for rock
discrimination. In this study, an RGB colour-composite was produced by assigning HV+HH
polarization channels to red, HV/HH polarization channels to green and HV polarization
channel to blue colour for detailed rock discrimination in the Kelantan state. Figure 7
represents the RGB merged image for the northern and southern parts of the Kelantan state.
Granitic rocks are manifested as light green colour (strong green in high altitude area
probably due to layover effect on SAR image (Gelautz et al., 1998; Franceschetti and Lanari,
1999)), while sedimentary and metamorphic rocks appears as a variety of tones such as
brown, light brown, dark green, light blue and light red in the image (Fig. 7). In comparison



with geological map of the study area, it seems that brown and dark green hues are
metamorphic rocks and light brown, light blue and light red colours are sedimentary rocks
(clay, shale, conglomerate, sandstone and limestone). River systems and lakes are observable
clearly as black meandering stream and water bodies especially in the northern segment of
the region (Fig. 7).

***5.3 Structural mapping at district scale and Landslide and flood risk delineation***

Four directionally filtered images of fine mode observation, which contain enhanced
information for set of lineaments in N-S, E-W, NE-SW and NW-SE direction, were used for
lineament mapping in the Kelantan state. Figure 8 shows structural map for the Kelantan
state, which is derived from the resultant image of directional filtering to HV polarization
channel. It should be noted that the locations of wet lands are also portrayed in this figure.
The most important structural features in the image map are fault zones, river systems and
drainage lines patterns (Fig. 8). Within the study area two large fault zones are presented,
which are the BRSZ in the north-west and Lebir Fault Zone in the south-east. The N-S, NE-
SW and NNE-SSW lineament trends are commonly dominant in the image map. The
dominant lineaments tend to run in the N-S direction, which is mainly linked to the N-S
trending of the BRSZ (in the west) and Lebir Fault Zone (in the east). Additionally, few short
NW-SE trending lineaments are detected in the western and eastern parts of the study area
(Fig. 8). The N-S and NE-SW striking system distributed in the south eastern segment of the
region is particularly related to Lebir Fault Zone. Pattern of the lineament map in north-
western part of the image map displays the occurrence of BRSZ fault zone, which contains
lineaments in N-S and NNE-SSW directions. Most of the short and smaller faults follow the
N-S, NE-SW and NNE-SSW trends as the major fault systems in the Kelantan state. It seems

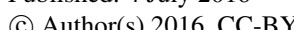

that the NW-SE faults are the youngest faults in the study area due to low frequency of
lineaments in this trend. The major N-S trending faults are interpreted as oldest structures in
Peninsular Malaysia and related to amalgamation of Gondwana-derived terranes during the
Permian-Triassic, and NE-SW and NW-SE-trending faults are interpreted to be Late Triassic
to Jurassic in age (Metcalfe, 2013b).
The occurrence and concentration of geological structural features in any region are related to
rock type, thickness of uppermost weathered mantle and brittleness of rocks (Harris et al.,
1960). In the Kelantan state, high concentrations of lineaments are associated with granitic
rocks in higher terrains and low concentrations of lineaments are usually associated with
lower terrains of metamorphic and sedimentary rocks. Tjia and Harun (1985) analyzed
regional structures of Peninsular Malaysia and reported that many lineaments are very well
displayed in areas underlain by granite, whereas in other areas they are poorly shown.
Moreover, Akhir (2004) identified that high lineament concentrations in the Upper Perak
Valley, Perak state, Peninsular Malaysia are closely related to higher terrain which is formed
by more resistant rocks such as granite and sandstone, whereas low lineament concentrations
are associated with lower terrain which is mainly formed by less resistant rock types such as
shale and slate.  As well, volcanic rocks in the Upper Perak Valley, Perak state encompass a
low lineament concentration (Akhir, 2004).
The rivers in the study area are structurally controlled. They display zigzag patterns due to
the presence of fractures, joints and faults with changes in orientation (Fig. 8). The drainage
system in the Kelantan river basin shows dendritic, sub-dendritic and rectangular patterns
(Fig. 8). It is evident that the drainage pattern is apparently being controlled by structure and
lithology in the study area. However, several factors such as topography, soil type, bedrock
type, climate and vegetation cover influence input, output and transport of sediment and
water in a drainage system. The geological structures and lithologic variation have given a



rise to different drainage patterns (Summerfield, 1991). For instance, dendritic and sub-
dendritic pattern with a large number of tributaries are typical of drainage in areas of
impermeable crystalline rock such as gneiss and/or sediment of uniform resistance
(horizontal strata). The pattern is characteristic of essentially flat-lying and/or relatively
homogeneous rocks and impervious soils with lack of structural control. Rectangular pattern
is usually caused by jointing or faulting of the underlying bedrocks. It is usually associated
with massive, intrusive igneous and metamorphic rocks (Summerfield, 1991). Therefore, it is
assumed that the area with dendritic and sub-dendritic pattern is subjected to hydro-
geological hazards such as flooding because of low infiltration runoff. Rectangular drainage
pattern is susceptible zone that could be easily affected by landslide due to slope of the land,
litho-structural conditions and speed of runoff. In the Kelantan river basin, most of the
dendritic and sub-dendritic drainage patterns are detected in central part of the river basin
(Fig. 8), which is consisted of sedimentary rocks. However, most of the rectangular drainage
patterns are associated with igneous and metamorphic rocks in the western part of the
Kelantan river basin (Fig. 8). Structural and topographical feature map of the Kelantan river
basin is shown in Figure 9. It is evident that most of the dendritic and sub-dendritic drainage
patterns are located in low lands and the rectangular drainage pattern is dominated in high
lands in the Kelantan river basin.

*5.4 Field observation*
Field observations were conducted between 20 and 25 June 2015 to compare the detected
high potential risk and susceptible zones with high damaged areas in recent flooding events in
the Kelantan river basin. GPS surveying was carried out in Tanah Merah, Machang, Jeli,
Kuala Krai and Gua Musang districts in the Kelantan river basin. 453 landslide affected
zones and flooded areas locations were recorded in forest, rubber, bushes (degraded forest),

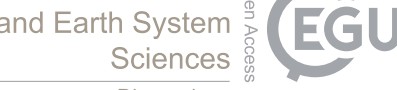



mixed crops, oil palm, cleared land, urban area and agriculture lands. Rock samples were
taken from the lithological units of landslide affected sites. Ground photographs were taken
of the high damaged areas after 2015 flooding event. The analysis of field investigations data
indicate that many of flooded areas were associated with high potential risk zones for hydro-
geological hazards such as wetlands, urban areas, floodplain scroll, meander bend and
dendritic and sub-dendritic drainage patterns. Most of the hydro-geological hazards zones are
located in flat topography regions (Fig. 9). Flat topography with impervious soils/surface has
low infiltration runoff, which yields more storm runoff during heavy monsoon rainfall
compared to other regions. Figure 10 (A and B) shows meander bend and floodplain scroll
zones in in the Kelantan river basin, which were flooded areas during 2015 flooding event.
Numerous landslides affected points were recorded in high-altitude segment of south and
south-western part of the Kelantan state. As mentioned above, high drainage density of
rectangular system is governed in this domain. The drainage density affects runoff, in that a
high drainage density drains runoff water rapidly, decreases the lag-time and increases the
peak of hydrograph. Consequently, the slope of the land in the south and south-western
regions increases the speed and extent of water and sediment transportation to the Kelantan
river basin during heavy monsoon rainfall. Most of the landslide affected points were located
in topographic slope of metamorphic and quaternary rock units. However, some landslides
occurred in fracture zones of weathered igneous rock units. Some large landslide affected
zones were recorded in the intersection of longitude and latitude between (N 5° 08′ 02″, E
101° 58′ 53″), (N 5° 08′ 14″, E 101° 59′ 06″), (N 5° 08′ 24″, E 101° 59′ 21″) and (N 5° 09′ 03″,
E 101° 59′ 41″). These landslide points were associated with N-S, NNE-SSW and NE-SW
trending fault zones. Figure 11 (A and B) shows large landslide affected zones in south-
western part of the Kelantan state.



## 6. Conclusions

Results of this investigation indicate that the PALSAR-2 onboard the ALOS-2 has proven to be successful advanced remote sensing satellite data for disasters monitoring in tropical environments. Analysis of the PALSAR-2 data provided significant information for identifying high potential risk and susceptible zones for natural hazards of geological origin in the Kelantan river basin, Malaysia. Wetlands, floodplain scroll, meander bend, dendritic and sub-dendritic drainage patterns and urban areas were identified as high potential risk zones for hydro-geological hazards. Landslide recurrence regions were detected in high-altitude segment of south and south-western part of the Kelantan state, which is dominated with high density of rectangular drainage pattern and topographic slope of metamorphic and quaternary rock units. Some of the large landslide zones were associated with N-S, NNE-SSW and NE-SW trending fault systems. Structural and topographical geology maps were produced for the Kelantan river basin that could be used to facilitate the planning of geo-hazards mitigation. In conclusion, the results of this investigation has great potential assistance in terms of total solution to flood disaster management in the Kelantan river basin by providing important source of information to assess the potential for many natural hazards of geological origin.

**Acknowledgements**

This study was conducted as a part of Transdisciplinary Research Grant Scheme (TRGS) (Vote no: R.J130000.7809.4L837), Ministry of Higher Education (MOHE), Malaysia. We are thankful to the Universiti Teknologi Malaysia for providing the facilities for this investigation.



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



**Figure captions**

Figure 1. Location of the Kelantan state in Peninsular Malaysia.

Figure 2. Geologic map of the Kelantan state (modified from Department of Minerals and Geoscience Malaysia, 2003).

Figure 3. RGB colour combination (HH, HV and HH+HV polarization channels) of PALSAR-2 Scan SAR scene covering northern part of the Peninsular Malaysia. Black rectangle shows covering area by fine mode merged images of the northern and southern parts of the Kelantan state in this study.

Figure 4. ScanSAR HV polarization image of the Kelantan state superimposed by general topography map.

Figure 5. ScanSAR image map derived from N-S (0$^{\circ}$), NE-SW (45$^{\circ}$), and NW-SE (135$^{\circ}$) directional filters for northern part of the Peninsular Malaysia. Explanation for the figure: dashed black lines = major faults; dot-dashed lines = folds and curvilinear; red lines = faults and fractures.

Figure 6. RGB colour-composite image of HH, HV and HH+HV polarization channels derived from fine mode merged image of the northern and southern parts of the Kelantan state.

Figure 7. RGB colour-composite image of HV+HH, HV/HH and HV polarization channels derived from fine mode merged image of the northern and southern parts of the Kelantan state.

Figure 8. Structural lineament map of the Kelantan state derived from directional filtering to HV polarization channel.

Figure 9. Structural and topographical feature map of the Kelantan river basin.

Figure 10. (A): Meander bend; and (B): floodplain scroll zone in the Kelantan river basin.





Figure 11. Large landslide affected zones (A and B) in south-western part of the Kelantan
state.

**Table captions**
Table 1: Directional filters with 5*5 kernel matrix.
Table 2: Directional filters with 7*7 kernel matrix.



















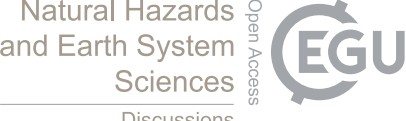

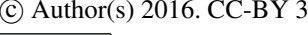




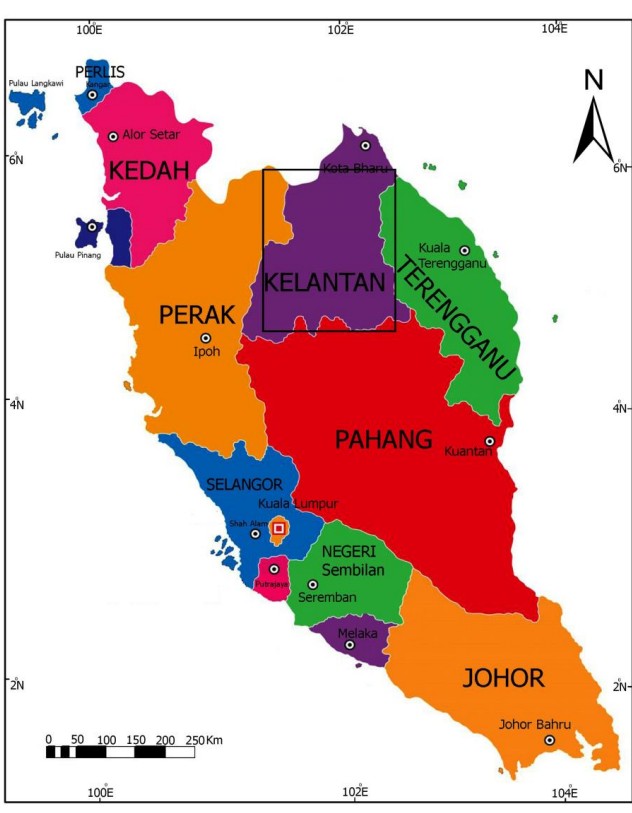


Figure 1. Location of the Kelantan state in Peninsular Malaysia.








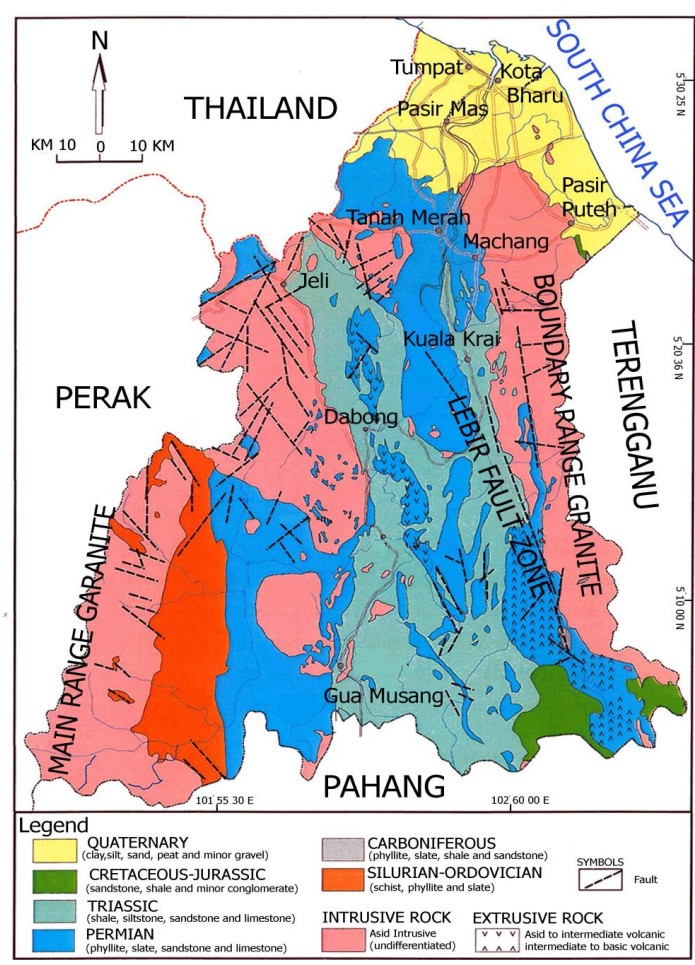


Figure 2. Geologic map of the Kelantan state (modified from Department of Minerals and
Geoscience Malaysia, 2003).










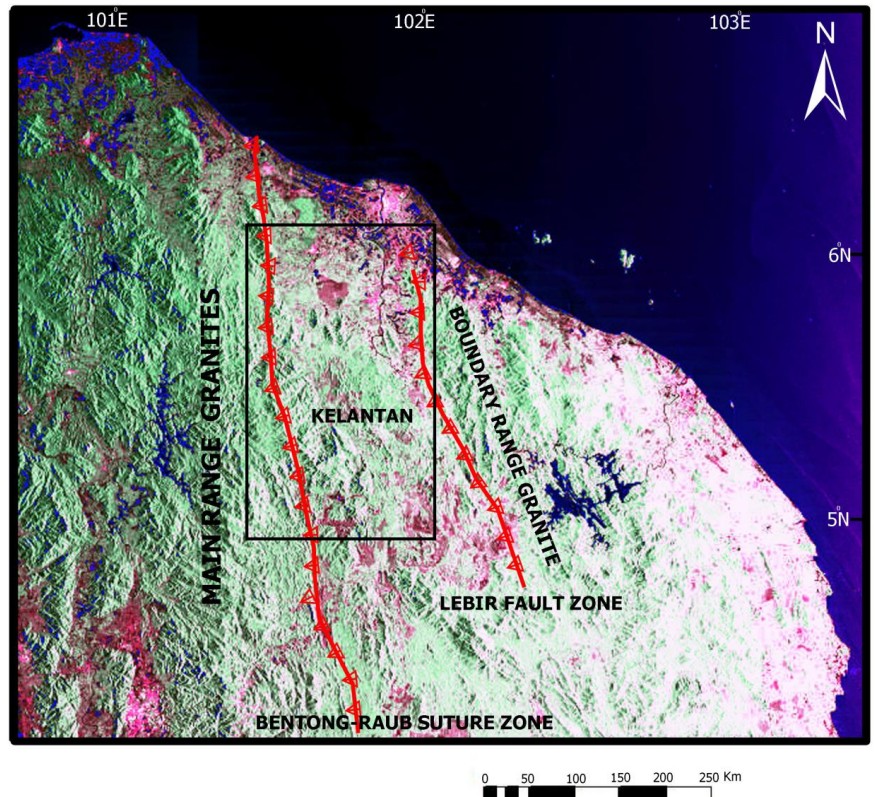

Figure 3. RGB colour combination (HH, HV and HH+HV polarization channels) of
PALSAR-2 Scan SAR scene covering northern part of the Peninsular Malaysia. Black
rectangle shows covering area by fine mode merged images of the northern and southern
parts of the Kelantan state in this study.











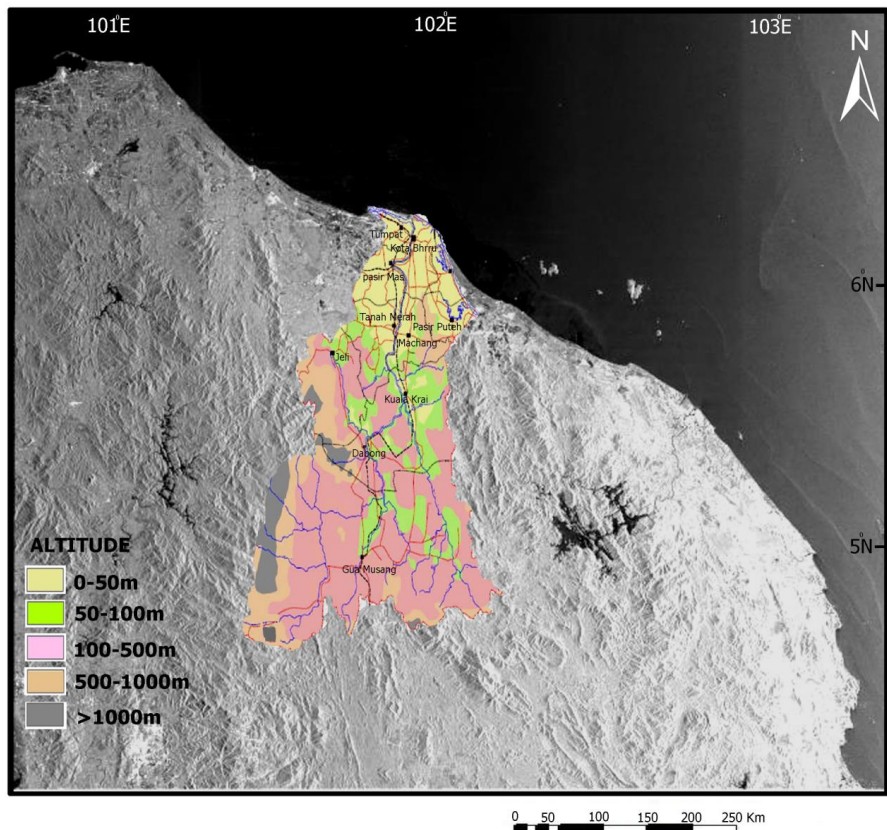



Figure 4. ScanSAR HV polarization image of the Kelantan state superimposed by general
topography map.









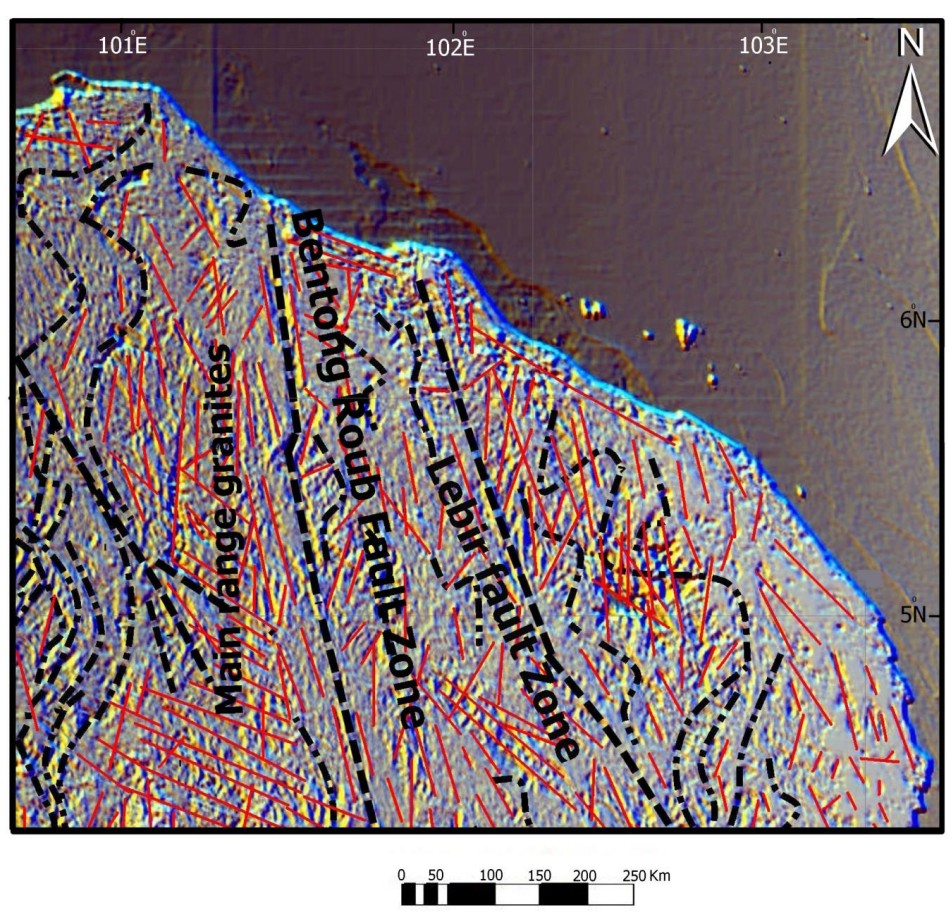

Figure 5. ScanSAR image map derived from N-S ($0°$), NE-SW ($45°$), and NW-SE ($135°$)
directional filters for northern part of the Peninsular Malaysia. Explanation for the figure:
dashed black lines = major faults; dot-dashed lines = folds and curvilinear; red lines = faults
and fractures.







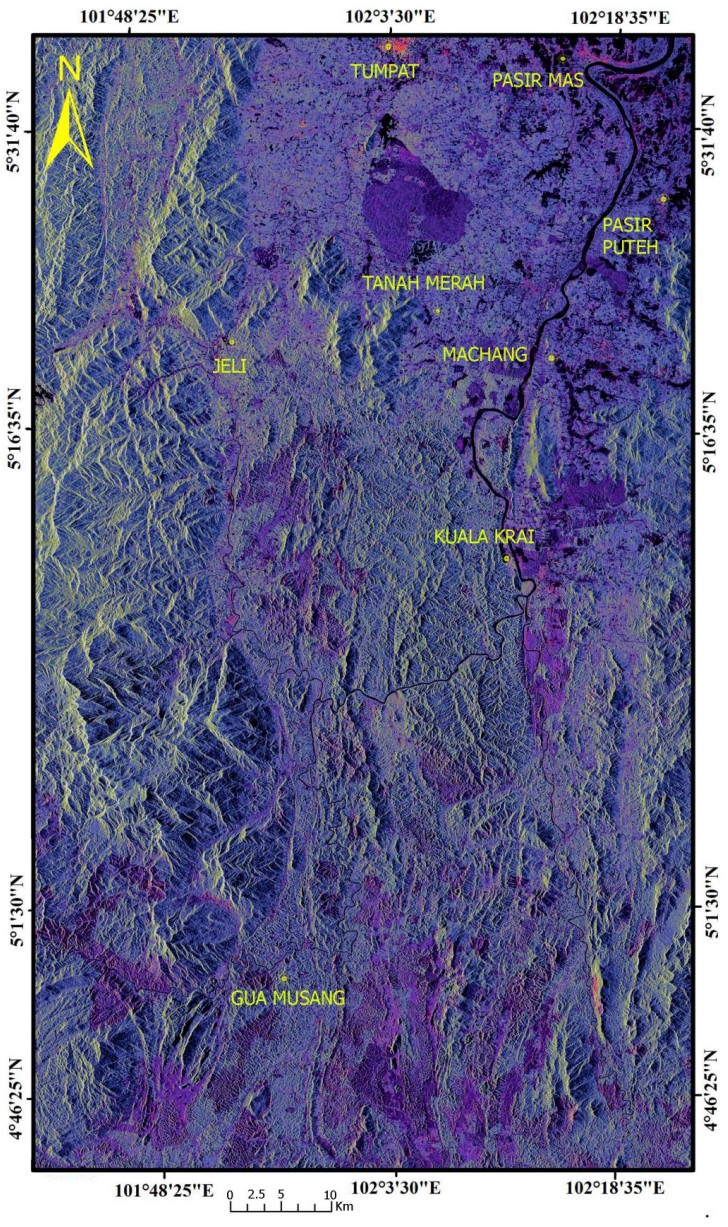



Figure 6. RGB colour-composite image of HH, HV and HH+HV polarization channels
derived from fine mode merged image of the northern and southern parts of the Kelantan
state.





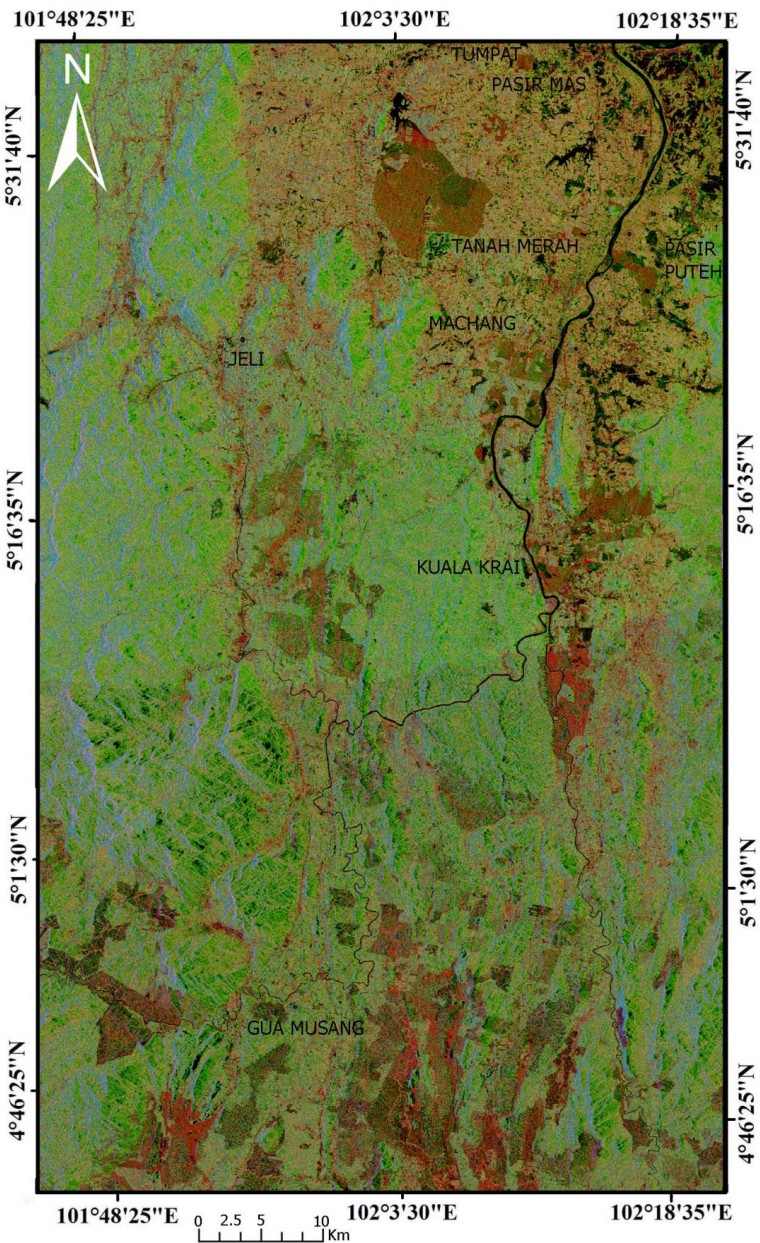

Figure 7. RGB colour-composite image of HV+HH, HV/HH and HV polarization channels
derived from fine mode merged image of the northern and southern parts of the Kelantan
state.





Figure 8. Structural lineament map of the Kelantan state derived from directional filtering to
HV polarization channel.

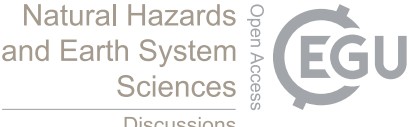





Figure 9. Structural and topographical feature map of the Kelantan river basin.






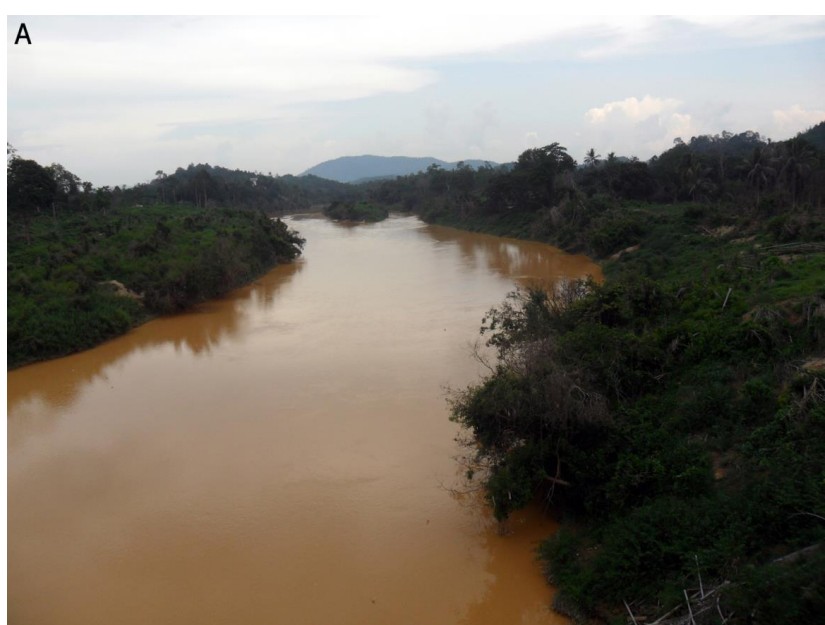


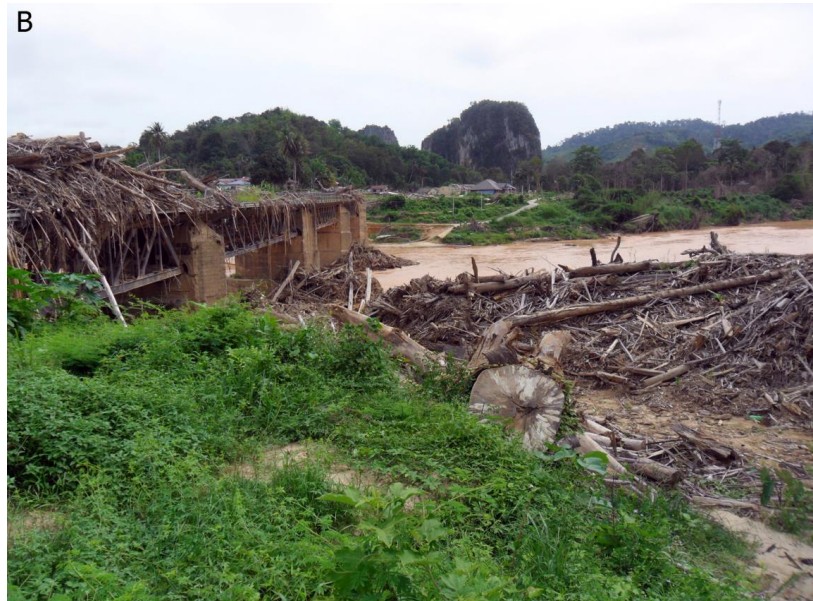

Figure 10. (A): Meander bend; and (B): floodplain scroll zone in the Kelantan river basin.






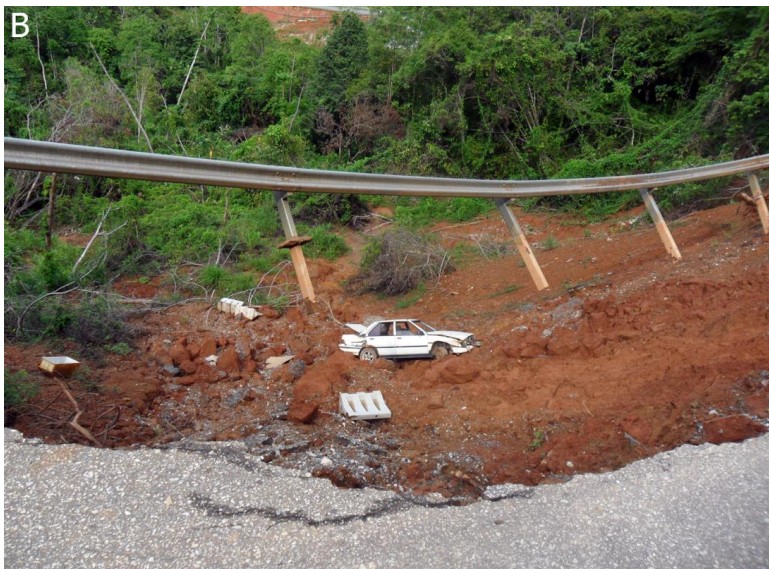


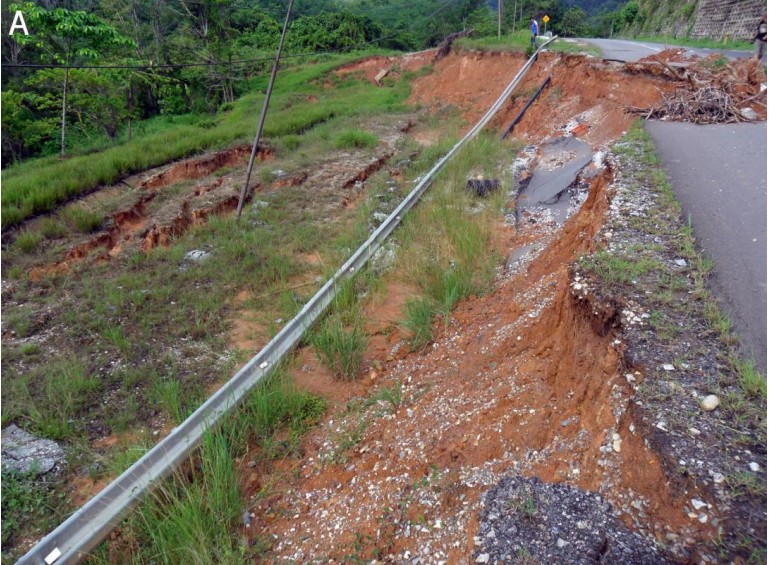

Figure 11. Large landslide affected zones (A and B) in south-western part of the Kelantan
state.







Table 1: Directional filters with 5*5 kernel matrix.

| N-S | | | | |
|---|---|---|---|---|
| -1.0000 | -1.0000 | 0.0000 | 1.0000 | 1.0000 |
| -1.0000 | -1.0000 | 0.0000 | 1.0000 | 1.0000 |
| -1.0000 | -1.0000 | 0.0000 | 1.0000 | 1.0000 |
| -1.0000 | -1.0000 | 0.0000 | 1.0000 | 1.0000 |
| -1.0000 | -1.0000 | 0.0000 | 1.0000 | 1.0000 |
| -1.0000 | -1.0000 | 0.0000 | 1.0000 | 1.0000 |
| -1.0000 | -1.0000 | 0.0000 | 1.0000 | 1.0000 |
| **E-W** | | | | |
| -1.0000 | -1.0000 | -1.0000 | -1.0000 | -1.0000 |
| -1.0000 | -1.0000 | -1.0000 | -1.0000 | -1.0000 |
| 0.0000 | 0.0000 | 0.0000 | -0.0000 | -0.0000 |
| 1.0000 | 1.0000 | 1.0000 | 1.0000 | 1.0000 |
| 1.0000 | 1.0000 | 1.0000 | 1.0000 | 1.0000 |
| **NE-SW** | | | | |
| -1.4142 | -1.4142 | -0.7071 | 0.0000 | 0.0000 |
| -1.4142 | -1.4142 | -0.7071 | 0.0000 | 0.0000 |
| -0.7071 | -0.7071 | 0.0000 | 0.7071 | 0.7071 |
| 0.0000 | 0.0000 | 0.7071 | 1.4142 | 1.4142 |
| 0.0000 | 0.0000 | 0.7071 | 1.4142 | 1.4142 |
| **NW-SE** | | | | |
| 0.0000 | 0.0000 | -0.7071 | -1.4142 | -1.4142 |
| 0.0000 | 0.0000 | -0.7071 | -1.4142 | -1.4142 |
| 0.7071 | 0.7071 | 0.0000 | -0.7071 | -0.7071 |
| 1.4142 | 1.4142 | 0.7071 | 0.0000 | 0.0000 |
| 1.4142 | 1.4142 | 0.7071 | 0.0000 | 0.0000 |





Table 2: Directional filters with 7*7 kernel matrix.

| N-S | | | | | | |
|---|---|---|---|---|---|---|
| -1.0000 | -1.0000 | -1.0000 | 0.0000 | 1.0000 | 1.0000 | 1.0000 |
| -1.0000 | -1.0000 | -1.0000 | 0.0000 | 1.0000 | 1.0000 | 1.0000 |
| -1.0000 | -1.0000 | -1.0000 | 0.0000 | 1.0000 | 1.0000 | 1.0000 |
| -1.0000 | -1.0000 | -1.0000 | 0.0000 | 1.0000 | 1.0000 | 1.0000 |
| -1.0000 | -1.0000 | -1.0000 | 0.0000 | 1.0000 | 1.0000 | 1.0000 |
| -1.0000 | -1.0000 | -1.0000 | 0.0000 | 1.0000 | 1.0000 | 1.0000 |
| -1.0000 | -1.0000 | -1.0000 | 0.0000 | 1.0000 | 1.0000 | 1.0000 |
| **E-W** | | | | | | |
| -1.0000 | -1.0000 | -1.0000 | -1.0000 | -1.0000 | -1.0000 | -1.0000 |
| -1.0000 | -1.0000 | -1.0000 | -1.0000 | -1.0000 | -1.0000 | -1.0000 |
| -1.0000 | -1.0000 | -1.0000 | -1.0000 | -1.0000 | -1.0000 | -1.0000 |
| 0.0000 | 0.0000 | 0.0000 | -0.0000 | -0.0000 | -0.0000 | -0.0000 |
| 1.0000 | 1.0000 | 1.0000 | 1.0000 | 1.0000 | 1.0000 | 1.0000 |
| 1.0000 | 1.0000 | 1.0000 | 1.0000 | 1.0000 | 1.0000 | 1.0000 |
| 1.0000 | 1.0000 | 1.0000 | 1.0000 | 1.0000 | 1.0000 | 1.0000 |
| **NE-SW** | | | | | | |
| -1.4142 | -1.4142 | -1.4142 | -0.7071 | 0.0000 | 0.0000 | 0.0000 |
| -1.4142 | -1.4142 | -1.4142 | -0.7071 | 0.0000 | 0.0000 | 0.0000 |
| -1.4142 | -1.4142 | -1.4142 | -0.7071 | 0.0000 | 0.0000 | 0.0000 |
| -0.7071 | -0.7071 | -0.7071 | 0.0000 | 0.7071 | 0.7071 | 0.7071 |
| 0.0000 | 0.0000 | 0.0000 | 0.7071 | 1.4142 | 1.4142 | 1.4142 |
| 0.0000 | 0.0000 | 0.0000 | 0.7071 | 1.4142 | 1.4142 | 1.4142 |
| 0.0000 | 0.0000 | 0.0000 | 0.7071 | 1.4142 | 1.4142 | 1.4142 |
| **NW-SE** | | | | | | |
| 0.0000 | 0.0000 | 0.0000 | -0.7071 | -1.4142 | -1.4142 | -1.4142 |
| 0.0000 | 0.0000 | 0.0000 | -0.7071 | -1.4142 | -1.4142 | - 1.4142 |
| 0.0000 | 0.0000 | 0.0000 | -0.7071 | -1.4142 | -1.4142 | -1.4142 |
| 0.7071 | 0.7071 | 0.7071 | 0.0000 | -0.7071 | -0.7071 | -0.7071 |
| 1.4142 | 1.4142 | 1.4142 | 0.7071 | 0.0000 | 0.0000 | 0.0000 |
| 1.4142 | 1.4142 | 1.4142 | 0.7071 | 0.0000 | 0.0000 | 0.0000 |
| 1.4142 | 1.4142 | 1.4142 | 0.7071 | 0.0000 | 0.0000 | 0.0000 |
