# Peer review of "Identification of high risk zones for geological origin hazards using PALSAR-2 remote sensing data: Kelantan river basin, Peninsular Malaysia"

_Natural Hazards and Earth System Sciences, 2016_

## Referee Comment (RC1) · Anonymous Referee #1 · 1 Aug 2016

In this study, the authors tried to identify and map high risk zones using ALOS PALSAR-2 data. The topic of this manuscript is quite interesting. However, the manuscript in this stage lasks innovation either from methodological or application aspects. Moreover, the results obtained in this study is suspected, such as Figure 3, 5 and 8. From my point of view, it is challenging to derive those maps by using simplied linear filters. If more processing details and mid-products can be shown, then the results could be more confirmative.

There are several concept misleadings: 1)Line 29-30, wetlands and urban areas can

not defined as geological hazards. 2)Line 199-200, HV polarization has a better penetration for the forest, but not subsurface. Thus, a more rigorous description is required to avoid the confusion.

---

## Author Comment (AC1) · 1 Aug 2016

A. Beiranvand Pour and M. Hashim

beiranvand.amin80@gmail.com

In this study, the results of the first attempt to use Phased Array type L-band Synthetic Aperture Radar-2 (PALSAR-2) data for the purposes of identification of high risk zones for geological origin hazards on tropical environments are presented for an area on Kelantan river basin, Peninsular Malaysia. This study included undertaking robust image processing and interpretation of a ScanSAR observation mode and two fine observation mode scenes associated with comprehensive fieldwork. Standard image processing methods were employed to extract geological information for mapping high

potential risk and susceptible zones for natural hazards of geological origin, which is broadly user-friendly approach for geological societies. This study is the first time that L-band SAR remote sensing data is used for identification of high potential risk and susceptible zones for natural hazards of geological origin in tropical environments. It is dire need to apply this approach in Malaysia and other parts of South East Asia that have inaccessible regions and high potential zones for natural hazards of geological origin, which are hidden by dense rainforest. This study and approach used is quite innovative and unique in the field of geological remote sensing.

For your information, this study is not a GIS study to show all stages of producing GIS layers and statistical models. Your suspecting is created because your background is different field. Normally, in the field of geological remote sensing we exhibit the final map (which contains several overlain layers) and we interpret the extracted remote sensing data based on our geological knowledge and existed references in the study area. For your information, there is variety of results using similar image processing technique to satellite remote sensing data for different geological study areas due to different aspects of local geology.

Line 29-30: (The analysis of field investigations data indicate that many of flooded areas were associated with high potential risk zones for hydro-geological hazards such as wetlands, urban areas, floodplain scroll, meander bend, dendritic and sub-dendritic drainage patterns, which are located in flat topography regions). Please read it carefully, we mentioned hydro-geological hazards! It is obvious for all readerships. However, many thanks for your concern.

Line 199-200: (HV polarization is more suitable for lineament extraction and edge enhancement in tropical environments than other polarization channels, because cross-polarization is more sensitive to lineament and also enhances penetration (Henderson and Lewis 1998; Pour and Hashim, 2015a,b). Penetration is proportional to wave-length, and cross-polarization also enhances penetration (Henderson and Lewis 1998).) It is obviously clear for all readerships, we do not mean subsurface. See

explanation as follows, which also could be found in introduction of this manuscript. Explanation: We are really thankful for your kind concern. Longer wavelengths optimize the depth of investigation of the radar signal and allow radar to have complete atmospheric transmission. Generally, the approximate depth of penetration is equal to radar's nominal wavelength. P-band shows the greatest penetration compared to the other bands. In particular, L-band microwave from PALSAR has ability to penetrate vegetation due to relatively long wavelengths (about 24 cm), making the data particularly useful for geological structural mapping in tropical environments (Igarashi, 2001; Rosenqvist et al., 2004; ERSDAC, 2006; Arikawa et al., 2010; Yamamoto et al., 2013; Pour and Hashim, 2013, 2014 a,b, 2015a,b; Shimada et al., 2015). The wavelength of the L-band is relatively long among microwaves (C-band: about 6 cm and X- band: about 3 cm), allowing it to travel all the way down to the ground through vegetation (Woodhouse, 2006). Not only can information be, obtained about vegetation but information of the ground surface can be obtained as well. Accordingly, L-band can observe the forest's underlying surface features as well as the canopy because of its penetration capability. Thus, in tropical environments, L-band SAR data is capable to provide the possibility of obtaining more useable geological structure information from the ground.

Please also note the supplement to this comment:
http://www.nat-hazards-earth-syst-sci-discuss.net/nhess-2016-191/nhess-2016-191-AC1-supplement.pdf

---

## Referee Comment (RC2) · Anonymous Referee #2 · 15 Aug 2016

Review Manuscript NHESS-2016-191

Identification of high-risk zones for geological origin hazards using PALSAR-2 remote sensing data: Kelantan river basin, Peninsular, Malaysia

The authors conducted a study using SAR data to map high-risk zones in a river basin in Malaysia. The information was compiled by visual interpretation.

General comments

Large parts of the manuscript have already been published in a conference paper from

which the authors copied text word by word. Even the figures and maps are taken from this publication: [1] A. Beiranvand Pour and M. Hashim, "Application of PALSAR-2 remote sensing data for landslide hazard mapping in Kelantan river basin, Peninsular Malaysia," Int. Arch. Photogramm. Remote Sens. Spatial Inf. Sci., vol. XLI-B8, pp. 413-416, 2016. The authors should at least take the effort to re-formulate the text. In addition it should be clear what distinguishes the new manuscript from the already published work (novelty aspect).

In general the manuscript needs some re-structuring to provide a logical information flow. Some redundant text should be removed (repetitive). The references used are overly outdated. In addition the manuscript needs English editing (use of article, grammar). See also detailed comments.

The study focused on landslides, which should be mentioned in the title in order to point out the content of the paper. This is also valid for the description of risk zoning within the text.

Self-citations (lines 47-48; lines 77-78): What is the sense of mentioning 5 (!) own papers in this sentence. Throughout the document the authors keep referring to their own papers. They should pick the most relevant publication or reason why all these publications are relevant. For example the reference Hashim et al. 2013 in line 52 is obsolete. There is plenty of literature before that referring to the limitations of optical remote sensing in the tropics due to cloud cover, e.g.: [1] R. Eckardt, C. Berger, C. Thiel, and C. Schmullius, "Removal of Optically Thick Clouds from Multi-Spectral Satellite Images Using Multi-Frequency SAR Data," Remote Sensing, vol. 5, p. 2973, 2013. [2] J. Ju and D. P. Roy, "The availability of cloud-free Landsat ETM+ data over the conterminous United States and globally," Remote Sensing of Environment, vol. 112, pp. 1196-1211, 3/18/ 2008. [3] F. Melgani, "Contextual reconstruction of cloud-contaminated multitemporal multispectral images," Geoscience and Remote Sensing, IEEE Transactions on, vol. 44, pp. 442-455, 2006.

[Figure]

Important scientific achievements in the mapping and analysis of landslides, hazards and disasters using SAR were omitted. The present manuscript uses RGB displays and filtering followed by visual interpretation. Other important information contained in SAR data, i.e. phase, coherence, InSAR, etc. were neglected. Relevant recent literature is not considered, e.g.: [1] S. Abuzied, S. Ibrahim, M. Kaiser, and T. Saleem, "Geospatial susceptibility mapping of earthquake-induced landslides in Nuweiba area, Gulf of Aqaba, Egypt," Journal of Mountain Science, vol. 13, pp. 1286-1303, 2016. [2] A. Barra, O. Monserrat, P. Mazzanti, C. Esposito, M. Crosetto, and G. Scarascia Mugnozza, "First insights on the potential of Sentinel-1 for landslides detection," Geomatics, Natural Hazards and Risk, pp. 1-10, 2016. [3] A. Bhattacharya, K. Mukherjee, M. Kuri, M. Vöge, M. L. Sharma, M. K. Arora, et al., "Potential of SAR intensity tracking technique to estimate displacement rate in a landslide-prone area in Haridwar region, India," Natural Hazards, vol. 79, pp. 2101-2121, 2015. [4] S. Bianchini, F. Raspini, A. Ciampalini, D. Lagomarsino, M. Bianchi, F. Bellotti, et al., "Mapping landslide phenomena in landlocked developing countries by means of satellite remote sensing data: the case of Dilijan (Armenia) area," Geomatics, Natural Hazards and Risk, pp. 1-17, 2016. [5] S. Plank, A. Twele, and S. Martinis, "Landslide mapping in vegetated areas using change detection based on optical and polarimetric SAR data," Remote Sensing, vol. 8, 2016. [6] F. Raspini, A. Ciampalini, S. Del Conte, L. Lombardi, M. Nocentini, G. Gigli, et al., "Exploitation of amplitude and phase of satellite SAR images for landslide mapping: The case of Montescaglioso (South Italy)," Remote Sensing, vol. 7, pp. 14576-14596, 2015. [7] R. Schlögel, J. P. Malet, P. Reichenbach, A. Remaître, and C. Doubre, "Analysis of a landslide multi-date inventory in a complex mountain landscape: The Ubaye valley case study," Natural Hazards and Earth System Sciences, vol. 15, pp. 2369-2389, 2015. [8] M. Vöge, R. Frauenfelder, K. Ekseth, M. K. Arora, A. Bhattacharya, and R. K. Bhasin, "The use of SAR interferometry for landslide mapping in the Indian Himalayas," in International Archives of the Photogrammetry, Remote Sensing and Spatial Information Sciences - ISPRS Archives, 2015, pp. 857-863. The authors should at least mention why these information were not used in the manuscript.

Scientific questions/issues (Specific Comments)

The description of the study area (lines 48-50) belongs to section 2 and not into the introduction.

What do the authors mean by "advanced remote sensing technology" (Line 63)? Please explain. It should be pointed out that the main information extraction was performed by visual interpretation.

The description of the ALOS systems could be shortened. E.g. remove details on ALOS-1 since the study only uses ALOS-2.

The statement "This study is the first time ..." doesn't hold. Using the keywords PALSAR, geology and hazards quite a number of publications can be found: [1] A. A. M. Radhi, M. L. R. Sarker, and N. Ishak, "Monitoring of surface deformation due to earthquake using dinsar technique and PALSAR-2 data: A case study of the Gorkha Earthquake in Nepal, 2015," in ACRS 2015 - 36th Asian Conference on Remote Sensing: Fostering Resilient Growth in Asia, Proceedings, 2015. [2] D. Notti, F. Calò, F. Cigna, M. Manunta, G. Herrera, M. Berti, et al., "A User-Oriented Methodology for DInSAR Time Series Analysis and Interpretation: Landslides and Subsidence Case Studies," Pure and Applied Geophysics, vol. 172, pp. 3081-3105, 2015. [3] X. T. Nguyen and C. P. Chang, "Mapping surface deformation in Red River Fault Zone using spaceborne SAR interferometry," in ACRS 2015 - 36th Asian Conference on Remote Sensing: Fostering Resilient Growth in Asia, Proceedings, 2015. [4] D. Sudiana, Rokhmatuloh, M. Rizkinia, Ardiansyah, R. Arief, B. Setiadi, et al., "Analysis of land deformation on slope area using PS InSAR. Case study: Malang area," in IOP Conference Series: Earth and Environmental Science, 2014. [5] R. F. Putri, S. Wibirama, I. Alimuddin, H. Kuze, and J. T. S. Sumantyo, "Monitoring and analysis of landslide hazard using dinsar technique applied to ALOS PALSAR imagery: A case study in Kayangan catchment area, Yogyakarta, Indonesia," Journal of Urban and Environmental Engineering, vol. 7, pp. 308-323, 2013. [6] N. Mitsuhara and T. Onuma, "Geological mapping in Iraq using

terra ASTER and ALOS PALSAR images and application to petroleum exploration," in 34th Asian Conference on Remote Sensing 2013, ACRS 2013, 2013, pp. 2051-2058. [7] Y. Yamada, "Mathematical morphology approach to detect farmland conditions from ALOS/PALSAR data after the 2011 off the pacific coast of Tohoku Japan earthquake and Tsunami," in International Geoscience and Remote Sensing Symposium (IGARSS), 2012, pp. 6118-6121. [8] K. Honda, T. Nakanishi, M. Haraguchi, N. Mushiake, T. Iwasaki, H. Satoh, et al., "Application of exterior deformation monitoring of dams by DInSAR analysis using ALOS PALSAR," in International Geoscience and Remote Sensing Symposium (IGARSS), 2012, pp. 6649-6652. [9] C. de Oliveira Andrades Filho and D. de Fáltima Rossetti, "Effectiveness of SRTM and ALOS-PALSAR data for identifying morphostructural lineaments in northeastern Brazil," International Journal of Remote Sensing, vol. 33, pp. 1058-1077, 2012. Please re-phrase.

Line 108-109 What do the authors mean by "largely lacking"? Does it mean there are no maps or only for part of the Kelantan river basin? Or are maps existing but not up-to-date?

From the text it is not clear what the objective (iii) Lines 115-116 is supposed to deliver. Please explain the purpose of objective (iii).

An overview on the manuscript is missing and should be inserted before Section 2.

Line 159-212 Is it really relevant to describe the entire PALSAR-2 system? Is the identification of the best imaging mode for the study purpose part of the study (Lines 172-173)? It is mentioned but not explicitly researched. What is the reasoning for the selected modes? The authors should consider to shorten these paragraphs and relate to the necessary information belonging to the study. The acquisition dates of the SAR data are missing and form a relevant information in the interpretation process, in particular for the link to the collected ground truth.

Remove "for comprehensive analysis of major ... of Kelantan." (Lines 178-179) This has been said before. Repetitive.

The description of precipitation and soil moisture and their influence on the data should be contained in the background information (introduction) and not in the description of the material section. Re-structure.

It would be useful to provide a kind of flowchart at the beginning of Section 4 to provide an overview on the image processing performed to the reader.

The authors should comment on the use of the median filter instead of considering specific speckle filters developed for polarized SAR images: [1] M. Xiaoshuang, S. Huanfeng, Z. Liangpei, Y. Jie, and Z. Hongyan, "Adaptive Anisotropic Diffusion Method for Polarimetric SAR Speckle Filtering," Selected Topics in Applied Earth Observations and Remote Sensing, IEEE Journal of, vol. 8, pp. 1041-1050, 2015. [2] S. Shitole, S. De, Y. S. Rao, B. Krishna Mohan, and A. Das, "Selection of Suitable Window Size for Speckle Reduction and Deblurring using SOFM in Polarimetric SAR Images," Journal of the Indian Society of Remote Sensing, vol. 43, pp. 739-750, 2015/12/01 2015. [3] L. Jong-Sen, T. L. Ainsworth, W. Yanting, and C. Kun-Shan, "Polarimetric SAR Speckle Filtering and the Extended Sigma Filter," Geoscience and Remote Sensing, IEEE Transactions on, vol. 53, pp. 1150-1160, 2015. [4] L. Gomez, M. E. Buemi, J. C. Jacobo-Berlles, and M. E. Mejail, "A New Image Quality Index for Objectively Evaluating Despeckling Filtering in SAR Images," Selected Topics in Applied Earth Observations and Remote Sensing, IEEE Journal of, vol. PP, pp. 1-11, 2015. [5] L. Fengkai, Y. Jie, and L. Deren, "Adaptive-Window Polarimetric SAR Image Speckle Filtering Based on a Homogeneity Measurement," Geoscience and Remote Sensing, IEEE Transactions on, vol. 53, pp. 5435-5446, 2015. In addition it is not clear if all the statements about the filtering process to enhance edges are valid for this particular case (ALOS PALSAR) or in general including optical imagery. The references regarding edge enhancement filters are very old (1995, 1996, 1997, 2000, 2008) - Lines 262-264. In the discussion of directional edge detection it is not obvious why all these references are cited: Haralick et al. 1987, Carr 1995, Sabins 1996, Vincent 1997, Jensen 2005). What are the particulars of these publications/research results? Haralick et al. 1987 is

the most cited document, which explains texture and feature detection. If the authors like to cite all the others too, it should have a reason.

Lines 291-296 The collected ground truth should be provided in a table or map to provide an overview on the content and distribution of the collected information.

Throughout the document there is no discussion on the choice of RGB assignments of the different channels. The authors should point out if they refer to standard channel assignments as used in geology or if the assignments were particularly applied for this study only. If the latter is the case it should be reasoned.

The information contained in lines 373-378 belongs to the background information (introduction). Similarly the text of lines 391 - 395 should be moved to the introduction.

It is not clear how the analysis of the field data has been done (Lines 480-483). Please clarify. Apparently major analysis efforts were done by visual interpretation. Looking at the application and the data that went into the analysis process it would be interesting to know why the authors did not consider using a GIS analysis and tools to provide rules for computer-supported analysis.

Line 505-507 What are "advanced remote sensing satellite data"?

Line 517-520 What is a "total solution" how do the results of this study relate to flood disaster management? The manuscript does not contain any research on flood management nor tools to do so. The work provides information from SAR data that could feed into a GIS.

Technical corrections

Line 61 "... between active faults, ..."

Line 62 "Therefore, the delineation of faults ..."

Line 63 "... in any region is a necessity to ..."

Lines 70-72 "ALOS-2 contains a Phased Array type L-band Synthetic Aperture Radar-2 (PALSAR-2) using microwaves to maximize its ability compared to ALOS-1, which contained three sensors, i.e. two optical and one microwave device."

Line 74 "... PALSAR-2 has the ability to ... due to its relatively long wavelength (..."

Lines 78-79 Repetition. Delete sentence.

Line 80 Complicated sentence: "Not only ... well." Re-phrase.

Line 86 "... increase in the amount of ..."

Line 92 "To date, only a few studies ..."

Line 96 The word 'dire' doesn't seem to be appropriate in this context. Suggestion: "There is an urgent need to apply ..."

Line 106 "A recent challenge ..."

Line 111-112 Remove explanation of PALSAR-2 "ALOS-2 Phased Array type L-band Synthetic Aperture Radar-2". The abbreviation has been defined in the beginning and should be used throughout the text. Remove "recently launched" (already explained in the beginning). Remove "remote sensing".

Line 120 "... composed of a central ... of the Southeast Asian ..."

Line 121 "... is located in the north-eastern ..."

Line 124 "The Kelantan river ..."

Line 125 "... of a flat slope ... in the northern ..." Line 137 "... Pahang states and Thailand (Fig. 2)."

Line 148 "... of Kelantan is divided into four ..."

Lines 190-193 "The HV channel ..." Avoid repetition. The information on available polarizations could be summarized in one sentence.

[Figure]

Line 194 "Particularly the HV channel ... modes increases the amount ..."

Line 200 "... proportional to wavelength, and ..."

Line 201 "Therefore, the HV polarization ..."

Line 230 "This flexibility makes convolution one of the most useful tools in image processing." - only for the purpose of feature extraction, in this case lineaments!

Line 233 Is this valid for all remote sensing images or does it relate to SAR?

Line 239 What are 'systematic image processing techniques"?

Line 242 "... images reduces the detectability ..."

Line 257 "In this study, a 3x3 neighborhood ..."

Line 258 "The Image Add Back used was 60%."

Line 262 "Edge enhancing filters highlight any ..."

Lines 265-268 Repetitive text. This has been stated above.

Line 267 "The directional filtering ..."

Line 273 "The directional filter ..."

Line 278 "A 5x5 kernel was selected for the ScanSAR scene ..."

Line 280 "A 7x7 kernel was applied to the Fine scenes." What are the references Chavez and Bauer 1982 and Jensen 2005 used for? It is not obvious from the context.

Line 282 "... were adjusted to N-S ..."

Line 284 "...  in counterclockwise direction." "An Image Add Back value of 60% was used."

Line 286 Please explain "systematic remote sensing techniques". It is not clear which images (polarizations) have been entered into the red, green and blue channels, respectively.

Line 304 "5.1 Lineament extraction and lithology ..."

Line 306 "The image acquired in ScanSAR mode was used ... structures. It shows mega-geomorphology ..."

Lines 308-310 "Figure 3 shows an RGB ... for the ScanSAR median filtered image.

Line 323 "... shows a ScanSAR ... by a general topography ..."

Line 326 The range '500-100 m' should be mentioned as "100-500 m". What is the meaning of differentiating 100-500 m and < 1000 m? Similarly Line 329 (range 50-500 m). This overlaps with 100-500 m. Please explain the categories.

Line 364 "Figure 6 shows the fine mode ..."

Line 365 "..., the HH polarization ..."

Line 367 "... an image map containing important ..."

Line 371-372 "Hence, the antenna did not receive a returned radar signal."

Line 374 Re-phrase: "Soil moisture/wetness information is best obtained from L-band radar because of its ground penetrating capability."

Line 379 "... distributed in the northern ..."

Line 382 "... observable in the central ..."

Line 393 "Hence the combination of ..."

Line 398 Is this really a merged image or a colour composite?

Line 403 "... with the geological map ..."

Line 411 "... images of the fine mode observations, ..."

Line 413 "... shows a structural ..."

Line 421 "... , a few short ..."

Line 424 "The pattern of ..."

Line 456 "The pattern is characterized of ..."

Line 457 "The rectangular pattern ..."

Line 461 "Rectangular drainage patterns form a susceptible zone ..."

Line 465 "..., which consists of ..."

Line 467 "The structural and ..."

Line 477 "... and flooded areas were ..."

Line 480 "... after the 2015 flood event." "... of the field data incdicates that ..."

Caption Figure 3 "... covering the northern ... The black rectangle shows the area covered by the fine mode RGB colour composite of the northern ..."

Caption Figure 5 "... filters for the northern part ..."

Caption Figure 6 "... from the fine mode images of the ..."

Caption Figure 7 "... derived from the fine mode images of the ..."

Caption Figure 8 This map was not derived from directional filtering but from visual interpretation of filtered imagery. Please correct.

———————————————————

---

## Short Comment (SC1) · 16 Aug 2016

P.nbsp;M. Marghany

magedupm@hotmail.com

This paper and previous work have published by authors are doubtful. The authors do not understand the mechanism of polarimetric SAR data processing. It seems that authors are experts in dealing with composite colour images which can be done by any software, including Photoshop.

The authors have claimed that L-band with different polarization is able to penetrate ground/soil which are located under heavy vegetation covers. How deep of L-band penetration with different polarization data? With this regard, authors are required to

model penetration depth with different polarization data. As I know this study area is located in tropical zone. So what is the effect of soil moisture on radar signal's penetration depth?

What types of speckles are contained with polarimetry data to be removed by directional filter?

How did authors avoid artefacts produced by directional filter? I think authors are required to show us mathematically the equation used with directional filter and how does it work with polarimetry SAR data?

Authors are not familiar with dealing with SAR data as they said that "Additionally, L-band is less affected by the growth of vegetation, which is useful for SAR interference analysis (Interferometry)." So what about TanDEM-X data?

In addition, authors are still proving them poor knowledge regarding dealing with SAR data processing as said" Compared to shorter wavelength SAR when some diastrophism takes place due to an earthquake or a volcanic activity and floods or landslides caused by a natural disaster "This sentence said the rest of SAR data are not accurate with dealing with nature disaster which is scientifically wrong. I think both authors are well practicing copy and paste by using rephrasing software as this common activity in this type of low class education and Research University.

Moreover, authors are claimed that they are only users of PALSAR-1/2 for geological mapping "To date, few studies used L-band SAR remote sensing data for Geological structure mapping in tropical environments (Pour and Hashim, 2013, 2014a,b, 2015a,b)." I went through these doubtful publications even are published in high impact factor journals, which are wrong scientifically as the gold is required geophysics equipment, chemical analysis and gold is located in deep ground with more than 40 m so how PALSAR will penetrate this depth?

I just like to show everyone how wrong scientifically paper can be published by help of

network or chief editor boards.

Although the authors have stated that high potential zones for natural hazards of geological origin hidden by dense rainforest but they cannot develop system or technique based on PALSAR data to detect geological features are hidden under rainforest.

The objectives of this study are ambiguities and cannot be achieved by using directional filter?

In section 3- Materials

The authors in spite of they are staying in tropical area so they do not know what the different between dry seasons and dry soil "Hence, SAR data acquired during dry seasons contain more useful information for detailed geological structural mapping in tropical environments".

Did the authors measure the soil moisture under heavy rainforest? In addition,

The authors claimed using cross-polarization. Can authors show the proof figure of cross-polarization?

Authors are required to prove what they stated "Therefore, HV polarization channel records more geological features that cover by dense vegetation".

Authors are required to show the low and high frequency details in PALSAR data in quantity forms. How mathematically as authors are dealing with mapping and polarimetry data processing, convolution filter can be used to detect low and high frequency in PALSAR data?

Authors are required to prove that medium filter has reduce the noise in polarimetric SAR data.

Authors are required to prove that directional filter can solve shortening and overlaying of features.

The authors are delivering cheating work : they said the data collected (June to August, 2015) and field data are collected out of this date. It means the ground data did not coincide with PALSAR-2 overpasses.

In addition, how the authors collected landslide data, I think this is required geotechnical equipment? How did the authors correlate rock data, with PALSAR-2 data?

How authors did collected 453 landslide affected locations and then detected them in PALSAR data?

How did authors use old data of "Department of Minerals and Geoscience Malaysia, 2003" with PALSAR-2 data 2015? As authors mentioned the study area was exposed to flood disaster during 2015.

Results section:

The work in polarimetry data cannot be produced by composite RGB colour which is false colour. This is required dealing with designing mathematical algorithms either for automatic or semi-automatic object detections in polarimetry data. It is clear that authors just draw manually by whatever image editing software. This type of work delivered wrong information for readers, students and experts

How did authors overlay old topography data with new palsar data even the time difference is 12 yrs. see Figure 4?

Figure 5 is output of directional filters, in this figure directional filter is produced wrong information as indicated in stripping zone of the image. Authors try to convince us wrongly by manual drawing (in red colour) that features are delivered by directional filter. Scan SAR is low resolution data so how directional filter can produce accurate features?

Figure 6 is fine mode data which contains a lot of polarimetric SAR problems which does not coincide with results of figure 7. It is clear that authors overlaid optical data with fine PALSAR mode data.

Figure 8 cannot be derived from directional filter as direction filter not same Hough and Canny algorithms which producing vector layers. I think the authors just overlaid the digitized vector topography map on fine mode palsar-2 data.

Another cheating procedures are very clear in Figure 9 as the authors overlaid the Figure 8 with a claimed DEM. It is not DEM, I think the authors have used colon table in ENVI to produce just fake DEM in fine PALSAR mode data.

In conclusion section, authors stated that "Results of this investigation indicate that the PALSAR-2 onboard the ALOS-2 has proven to be successful advanced remote sensing satellite data for disasters monitoring in tropical environments" what is the proof? I do not see any proof has shown through the output figures!

In addition, authors concluded that "In conclusion, the results of this investigation has great potential assistance in terms of total solution to flood disaster management in the Kelantan river basin by providing important source of information to assess the potential for many natural hazards of geological origin. " However authors never provide any detection of flooded area in PALSAR-2 data!

Finally, I do not advise publishing this wrong scientifically work in polarimetric SAR data. The novelty of polarimetric SAR data theory is absent in addition to the novelty of image processing procedures have been used in this work.

---

## Author Comment (AC2) · 25 Aug 2016

A. Beiranvand Pour and M. Hashim

beiranvand.amin80@gmail.com

We really appreciate your very useful and constructive comments.

General comments:

The mentioned conference manuscript has been published in early August (after the date of NHESS interactive discussion (4 July 2016)), which derived from primary results of our research around 8 month ago. I did not attend and register to ISPRS congress 2016 (23rd International Archives of the Photogrammetry, Remote Sensing and Spatial Information Sciences Congress, ISPRS 2016Í¿ PragueÍ¿ Czech RepublicÍ¿ 12 July
2016 through 19 July 2016Í¿ Code 122461), because I was chairman in IGARSS 2016, Beijing, China. I assumed that the ISPRS congress 2016 never published my very primary results of the similar research. However, NHESS Editor has applied similarity index before accepting for NHESS interactive discussion, so if there was any high similarity index, based on their policy the manuscript could not be accepted for interactive discussion. Accordingly, based on your comment we will re-formulate and reformat the text of manuscript after receiving editor permission for revision. We will add new data derived from Landsat-8 and several GIS layers using AHP approach were produced recently for balancing between our visual interpretation and statistical sector of the manuscript (please check attached file). Landslide susceptibility map were produced for the study area. Manuscript will be reconstructed, edited and redundancy will be removed. References based on your provided literatures will be updated. Landslide will be added to the revised title for more clarity for readership.

Specific comments: we will apply all your constructive comments to revised manuscript after finishing interactive discussion and permission form editor for revision. Please check follows to see we apply your comments to reconstruct a new manuscript.
* * *
In this investigation for applying AHP approach, 10 factors such as slope, aspect, soil, lithology, NDVI, land cover, distance to drainage, precipitation, distance to fault, and distance to road were extracted from Landsat-8 images, PALSAR data and fieldwork as shown in Figure ....

[Figure]

Figure .... List of extracted factors from from Landsat-8, PALSAR, and fieldwork. A: Slope (degree), B: Aspect (degree), C: Lithology types, D:Landuse type, E:Precipitation (mm), F: NDVI, G: Distance to main road, H: Distance to main river (m), and I: Distance to fault line (km).

**Fig. 1.**

Figure ... Methodology flowchart to produce landslide susceptibility map using AHP approach.

**Fig. 2.**

Figure … Distribution of various sizes of landslides identified through satellite-based detection and field observations after the flood episode in Kelantan 2014.

**Fig. 3.**

Figure …. Spatial distribution of landslide susceptibility map of each class before and after flood episode in Kelantan 2014 where class A: low risk, B: moderate risk, C: fair risk, D: high risk and E: very high risk.

**Fig. 4.**

[Figure]

Figure …. LSM before (A) and after (B) 2014 flooding episode in Kelantan state.

Figure ... Landslides occurrences at different land use type in Kelantan state.

**Fig. 5.**

Table ... Pair-wise comparison matrix for each input of AHP.

| Factor | Slope | Aspect | Soil | Lithology | NDVI | Land Cover | Precipitation | Dist to Road | Dist to River | Dist to Fault |
|---|---|---|---|---|---|---|---|---|---|---|
| Slope | **0.032** | 0.031 | 0.029 | 0.018 | 0.030 | 0.034 | 0.023 | 0.060 | 0.014 | 0.014 |
| Aspect | 0.032 | **0.031** | 0.036 | 0.070 | 0.030 | 0.021 | 0.015 | 0.050 | 0.014 | 0.014 |
| Soil | 0.161 | 0.123 | **0.145** | 0.175 | 0.075 | 0.206 | 0.226 | 0.150 | 0.170 | 0.211 |
| Lithology | 0.065 | 0.015 | 0.029 | **0.035** | 0.050 | 0.026 | 0.015 | 0.060 | 0.019 | 0.021 |
| NDVI | 0.161 | 0.154 | 0.291 | 0.105 | **0.150** | 0.206 | 0.090 | 0.100 | 0.226 | 0.211 |
| Land Cover | 0.097 | 0.154 | 0.073 | 0.140 | 0.075 | **0.103** | 0.135 | 0.100 | 0.170 | 0.169 |
| Precipitation | 0.065 | 0.092 | 0.029 | 0.105 | 0.075 | 0.034 | **0.045** | 0.060 | 0.019 | 0.021 |
| Dist to Road | 0.161 | 0.185 | 0.291 | 0.175 | 0.499 | 0.309 | 0.226 | **0.300** | 0.283 | 0.211 |
| Dist to River | 0.129 | 0.123 | 0.048 | 0.105 | 0.037 | 0.034 | 0.135 | 0.060 | **0.057** | 0.085 |
| Dist to Fault | 0.097 | 0.092 | 0.029 | 0.070 | 0.030 | 0.026 | 0.090 | 0.060 | 0.028 | **0.042** |
| **SUM** | 1.000 | 1.000 | 1.000 | 1.000 | 1.000 | 1.000 | 1.000 | 1.000 | 1.000 | 1.000 |

Table ... Factor weights for each input of AHP.

| Factors | Weight |
|---|---|
| F1 : Precipitation | 0.141 |
| F2 : Slope | 0.123 |
| F3 : Soil | 0.121 |
| F4 : Aspect | 0.102 |
| F5 : Lithology | 0.097 |
| F6 : Land Cover | 0.086 |
| F7 : Distance to Road | 0.084 |
| F8 : Distance to Drainage | 0.081 |
| F9 : NDVI | 0.073 |
| F10 : Distance to Fault | 0.062 |

**Fig. 6.**

---

## Author Comment (AC3) · 1 Sep 2016

Dear Dr. Rosa Lasaponara

I am writing to notify you about abusive comments which contain personal insults by M. Marghany to Nat. Hazards Earth Syst. Sci. Discuss., doi:10.5194/nhess-2016-191-SC1, 2016. Actually, he has introduced himself as Prof. Majed M. Marghany, but he has not promoted to professorship, yet. He was our ex-colleague and co-author, which has been terminated from our institute (Geoscience and Digital Earth Centre (Geo-DEC), Universiti Teknologi Malaysia) because of his mental problems and non-respectful behavior to office time and working days. He is not expert in field of geological remote sensing and landslide mapping and does not have any basic acknowledge understanding content of the manuscript. All of his abusive comments are not scientific because he did not read the manuscript properly. His abusive comments insulting to University Technology Malaysia '' I think both authors are well practicing copy and paste by using rephrasing software as this common activity in this type of low class education and Research University", which as really non-professional and academic. In addition, he insulted to our previous publications that are published in high standard and prestigious journals such as Ore Geology Reviews, Advances in Space Research, Remote Sensing and etc. ''I went through these doubtful publications even are published in high impact factor journals, which are wrong scientifically. I just like to show everyone how wrong scientifically paper can be published by help of network or chief editor boards ''.

His abusive comments could be seen in all written sentences. Accordingly, we respectfully request for censoring his comments that are not of substantial nature of direct relevance to the issues raised in the discussion paper and contain personal insults.

With Kind Regards
* * *